# A Pixel Design of a Branching Ultra-Highspeed Image Sensor

**DOI:** 10.3390/s21072506

**Published:** 2021-04-03

**Authors:** Nguyen Hoai Ngo, Kazuhiro Shimonomura, Taeko Ando, Takayoshi Shimura, Heiji Watanabe, Kohsei Takehara, Anh Quang Nguyen, Edoardo Charbon, Takeharu Goji Etoh

**Affiliations:** 1College of Science and Engineering, Ritsumeikan University, 1-1-1 Noji-Higashi, Kusatsu, Shiga 525-8577, Japan; skazu@fc.ritsumei.ac.jp (K.S.); tando@fc.ritsumei.ac.jp (T.A.); 2Graduate School of Engineering, Osaka University, 2-1 Yamada-oka, Suita, Osaka 565-0871, Japan; shimura@prec.eng.osaka-u.ac.jp (T.S.); watanabe@prec.eng.osaka-u.ac.jp (H.W.); 3School of Science and Engineering, Kindai University, 3-4-1 Kowakae, Higashi-Osaka, Osaka 577-8502, Japan; takehara@civileng.kindai.ac.jp; 4School of Electronics and Telecommunications, Hanoi University of Science and Technology, 1 Dai Co Viet, Bach Khoa, Hai Ba Trung, Hanoi 100803, Vietnam; quang.nguyen.anh@hust.edu.vn; 5Advanced Quantum Architecture Laboratory, EPFL, Rue de la Maladière 71b, Case Postale 526, CH-2002 Neuchâtel, Switzerland; edoardo.charbon@epfl.ch

**Keywords:** ultrahigh-speed imaging, branching image sensor, super-temporal resolution

## Abstract

A burst image sensor named Hanabi, meaning fireworks in Japanese, includes a branching CCD and multiple CMOS readout circuits. The sensor is backside-illuminated with a light/charge guide pipe to minimize the temporal resolution by suppressing the horizontal motion of signal carriers. On the front side, the pixel has a guide gate at the center, branching to six first-branching gates, each bifurcating to second-branching gates, and finally connected to 12 (=6×2) floating diffusions. The signals are either read out after an image capture operation to replay 12 to 48 consecutive images, or continuously transferred to a memory chip stacked on the front side of the sensor chip and converted to digital signals. A CCD burst image sensor enables a noiseless signal transfer from a photodiode to the in-situ storage even at very high frame rates. However, the pixel count conflicts with the frame count due to the large pixel size for the relatively large in-pixel CCD memory elements. A CMOS burst image sensor can use small trench-type capacitors for memory elements, instead of CCD channels. However, the transfer noise from a floating diffusion to the memory element increases in proportion to the square root of the frame rate. The Hanabi chip overcomes the compromise between these pros and cons.

## 1. Introduction

In 1991, Etoh developed a video camera with a parallel readout high-speed image senor of 4500 fps [1]. In response to requests from users of this camera for a much higher frame rate, Etoh and Takehara proposed an image sensor concept with in-pixel multiple analogue storage elements, targeting 30 Mfps in 1992 [2]. The paper also proposed an in-pixel video trigger concept which automatically stops a continuous overwriting operation upon a sudden change in a captured image. In 1996, Kosonocky et al. achieved 0.5 Mfps with an image sensor with in-pixel series-parallel-series (SPS) CCD storage [3]. He named such an image sensor with in-pixel storage for ultra-highspeed imaging “a burst image sensor”. In 2001, Etoh et al. reached 1 Mfps for the first time with an in-situ storage image sensor with a slanted linear CCD for each pixel [4]. In 2018, Etoh et al. developed a multi-framing camera with the frame interval Δt=10 ns (equivalent to 100 Mfps) [5]. When a chip with pixel-based driver circuits is stacked on the sensor chip with a 3D-stacking technology, the frame rate will reach 1 Gfps (Δt=1 ns) [6]. Kuroda et al. achieved 1.25 Gfps (Δt=8 ns) with a prototype CMOS burst image sensor. The sensor can store a large number of frames when the 3D-stacking is applied [7]. It took 26 years to go beyond the frame rate of 30 Mfps initially targeted in 1992.

The theoretical highest frame rate of silicon image sensors is 90.1 Gfps (Δt=11.1 ps) [8]. Recently, Ngo et al. proposed an image sensor concept with a germanium photodiode to achieve a frame rate higher than the theoretical limit of silicon image sensors. They named it super-temporal resolution (STR) image sensor [9]. Burst image sensors and STR imaging technologies achieved by other scientists are summarized in literature [9,10,11,12], including several historically important achievements [13,14,15,16,17,18,19,20].

A fundamental problem of burst image sensors is a trade-off between pixel count and frame count. Since the imaging area of an image sensor is fixed by standard formats, a large number of in-pixel storage elements for a large frame count increases the size of a pixel, reducing the pixel count. A 3D-stacking technology enables many storage elements in a stacked memory chip. However, a signal transfer rate from the sensor chip to the memory chip has another trade-off, this time, with signal transfer noise, since the noise level is proportional to the square root of the transfer rate.

To mitigate these tradeoffs, we propose a backside-illuminated (BSI) CCD/CMOS hybrid branching image sensor called Hanabi. The sensor has radially branching CCD channels and multiple CMOS readout circuits surrounding the channels as shown in Figure 1.

Hanabi will bring innovations to advanced scientific measurement technologies, including fluorescence lifetime imaging microscopy (FLIM), and imaging time-of-flight mass spectrometry (imaging TOF MS). In a pump-probe analysis, for example, an ultra-short laser pulse is applied to a specimen to excite electron states, and a phenomenon triggered by the laser shot is probed with a certain delay. Repeating the measurement for a stepwise change of the delay provides the temporal change of the phenomenon. The temporal resolution decreases to a femtosecond, depending on the width of the laser pulse. Hanabi will enable the measurement of the temporal change by one shot without repetition with the temporal resolution down to sub-nanoseconds.

This paper presents a feasibility study of the following aspects of Hanabi:(1)fundamental structure and operation schemes,(2)design conditions and criteria to evaluate basic performance such as frame rate, spatio-temporal crosstalk, and collision rate against walls of the guide pipe which is explained later.(3)evaluation of the basic performance related to the guide pipe and the front-side circuit layer, separately, by potential and Monte Carlo simulations,(4)overall performance for the combined structure of the guide pipe and the circuit layer.

The paper will also discuss super-temporal-resolution in Hanabi with a germanium photodiode, and finally summarize major contributions of this analysis.

## 2. Overview of Branching Image Sensor

### 2.1. Structure

#### 2.1.1. Guide Pipe and Circuit Layer

Figure 1a shows the cross section of Hanabi, which is composed of an upper square silicon pipe, receiving light incident to the backside, and a lower circuit layer on the front side.

Figure 1b shows the configuration of the elements on the front side. The sensor has multiple CCD branching gates radially arranged from the center of the pixel.

The planar shapes consist of a square for the guide pipe and a hexagon for the branching gates. It is easier to design a sensor composed of only square or hexagonal shapes. However, a preliminary analysis showed the advantages of the combined square and hexagonal structures as follows:(1)the hexagonal branching gates allow design of many storage elements in a proper balance of the areas of the gates, and(2)when the crystal orientation of the backside surface is (100), proper etching creates (100) surfaces on each of the walls of the square guide pipe.

While the configuration with mixed shapes makes the design complicated, we gave priority to better performance over saving the labors for the design.

The cascade branching begins from a guide gate (GG), hexa-furcating to first-branching gates (FG), each bifurcating to second-branching gates (SG), each connected to a floating diffusion (FD) for storage and readout of signal electrons, and further connected to a reset gate (RS) and a reset drain (RD).

The gates (electrodes) are systematically named as shown in Figure 1b. The first-branching gates are named FG_1_ to FG_6_, and also named FGs as a whole. Second-branching gates connected to FG_1_ are named SG_11_ and SG_12_; the first digit refers to the FG to which the second-branching gates are connected, and the second digit is numbered clockwise. Second-branching gates are named SGs as a whole. Likewise, the floating diffusions (FDs) and reset gates (RSs) are numbered. However, for burst imaging, numbering RSs is not necessary since signals are reset at once.

Figure 2 shows the cross-sections of BSI image sensors with a functional circuit layer on the front side which we previously proposed in [5]. Three technologies, a charge collection pyramid, a light/charge guide pipe, and a p-well have been proposed for (1) collection of signal electrons to the center of the pixel and (2) potential separation of the upper photodiode layer and the lower functional circuit layer. Hereafter, a light/charge guide pipe and a charge collection pyramid are shortened to a guide pipe and a pyramid.

The p-well separation was applied to the development of the following:(1)a burst image sensor which achieved the frame interval of 10 ns, and(2)an image signal accumulation image sensor (ISAS) which captures 1220 consecutive images at the frame interval of 40 ns, and can accumulate image signals by repeating image capturing of reversible events with very low incident light [20].

The guide pipe and the pyramid significantly increase frame rates [5], whereas the circuits under the structures are the same. The pyramid has a fill factor of 100% without an on-chip micro-lens. However, the process of the image sensor with the pyramid is still under development. On the other hand, the guide pipe can be made with existing process technologies. Therefore, in this paper, the guide pipe is employed to mainly study the feasibility and the performance of the multiple branching structures.

In this study, the guide pipe is surrounded by a silicon dioxide insulation layer. The outside can be crystal silicon, polysilicon, or other suitable materials.

To avoid collision of signal electrons against the walls of the guide pipe, a layer near each wall is to be negatively charged relative to the potential value in the bulk of the guide pipe. Several methods are available for creating the negative potential layer, including:(1)a thin boron-doped layer over the walls of the guide pipe,(2)a low voltage applied to the outside of the guide pipe, and,(3)a negatively charged thin layer over the insulation layer, made with Al_2_O_3_.

In this paper, a boron layer is employed. Some electrons still collide against the wall. Most of them are reflected to the inside, but some are trapped on the wall and later released or recombined with holes. The number of activated boron ions per wall area (the concentration multiplied by the thickness) was decided to reduce the collision probability to an acceptable level or better as shown later. In this paper, it is assumed that an electron once colliding with the wall is absorbed, which is a safe assumption for a conservative design.

#### 2.1.2. Equivalent Travel Routes of Signal Electrons

Let a collecting FD_C_ and waiting FD_W_s respectively denote the FD which collects signal electrons and other FDs waiting for their turn. The sensor includes twelve SGs and twelve FDs. Most of signal electrons arrive at GG. However, some fall on FGs and SGs due to the random motion. All these electrons flow down from eleven low-voltage (upstream) SGs toward a high-voltage (downstream) SG, and finally to FD_C_. Therefore, 132 (=12×11) electron travel routes form the upstream SGs to the FD_C_. However, if the planar pixel configuration is symmetric, all the routes are represented by one equivalent route. For the combination of the square guide pipe and the hexagonal gate, the number of equivalent routes increases as shown in Figure 3:
(1)from the upstream SGs to GG: there are three representative routes, R_A_, R_B_ and R_C_ for electrons to converge to GG, as color-coded in Figure 3b; for example, the green routes from SG_21_ (R_B_), SG_32_, SG_51_ and SG_62_ to GG are equivalent, since all these routes start from the corner of the guide gate, where the numbers of the gates are shown on FDs in Figure 3a, and(2)from GG to FD_C_ through an FG and an SG downstream: there are three representative routes, R_D_, R_E_ and R_F_, for the collected electrons to verge, as shown in Figure 3c.

Therefore, there are nine (=3×3) different equivalent routes. As shown later, slight differences among the downstream representative routes causes a serious difference on arrival times of electrons.

### 2.2. Operation Schemes

The sensor has two operation modes, burst imaging and continuous imaging. In burst imaging mode, upon photon detection, signal electron packets are stored in twelve FDs, which are converted to voltage signals and read out of the sensor or transferred to a stacked memory chip all at once after completion of the capture of a finite sequence of frames. In this mode, the frame count is only 12, however, one can bin multiple neighboring pixels to make a larger ‘macro pixel’ with the objective of increasing the frame count. For example, an interlace operation, in which image signals are alternately captured with even-column pixels and odd-column pixels doubles the frame count to 24, sacrificing the pixel resolution and the sensitivity in half. And macro pixel with four adjacent pixels quadruplicates the pixel count to 48 and makes the resolution 1/4.

The burst imaging mode has the following advantages:(1)when the sensor is cooled, the buried CCD channels theoretically enable a perfect and noiseless signal transfer even at ultrafast signal transfer, if the field is less than the critical field at which the drift velocity is saturated, and(2)noiseless signal accumulation is possible for repetitive burst imaging of reproducible events emitting very weak light:

Alternatively, continuous imaging mode by Hanabi with a stacked memory chip has the following advantages:
(1)a large frame count, and(2)the transfer rate of signal electrons from each FD to in-situ storage on the stacked chip can be reduced by 1/12 of that of an image sensor with one FD per pixel, reducing the transfer noise by 1/$\sqrt{12}$.

However, a large number of wires are necessary to independently operate FDs and reset gates in addition to the CCD electrodes. The number of wires for the reset gates reduces in alternate continuous imaging, in which image signals stored in the twelve FDs on the odd-number columns are simultaneously transferred to the stacked memory chip during successive image capturing by the pixels on the even-number columns, and vice versa. In a pixel, a similar operation can be introduced by grouping the FDs.

### 2.3. Design Conditions

#### 2.3.1. Theoretical and Practical Temporal Resolutions

We proposed the definitions of the theoretical temporal resolution $t_{T}\;$and the practical temporal resolution $t_{P}$, assuming a Gaussian distribution for the arrival time distribution of signal electrons to a detection plane. They are defined by means of the standard deviation σ of the distribution, an average penetration depth δ of incident light with a certain wavelength to a photodiode material, and the length of the photodiode L.

The theoretical temporal resolution is defined as $t_{T}=2\sigma \;$for L=δ [8,9]. For the first condition, two peaks of a sum of ordinates of two Gaussian distributions are combined to one peak with no dip at the center. Therefore, it is impossible to separately detect the signal electron packets which are generated by two pulses of light with the time difference of$\;t_{T}$ incident to the backside, and spread by diffusion and mixing toward the detection plane. The second condition, L=δ, means that the thickness of the photodiode is equal to the average penetration depth.

However, for these conditions, overlap of the two Gaussian distributions results in a large temporal crosstalk, and 36.8% ($=e^{-1}$) of incident light passes through the front side, and is wasted.

On the other hand, the practical temporal resolution is defined as $t_{P}=3.3\sigma$ for L=3δ. When the arrival time distribution is approximated by the Gaussian distribution, the first condition is equivalent to $t_{P}^{\prime }=t_{95}-t_{5},$ shown in Figure 4, where $t_{95}$ and $t_{5}$ are respectively the times when 95% and 5% of generated signal electrons arrive at the detection plane. Therefore, the temporal crosstalk is 5%. For L=3δ, 95% of the incident light is converted to signal electrons. In this paper, the practical temporal resolution $t_{P}^{\prime }$ is employed to evaluate the sensor design.

#### 2.3.2. Wavelengths of Incident Light and Fundamental Shape Parameters

The wavelengths of blue, green, and red light are assumed to be 450,550 and 650 nm. The wavelength of 550 nm for green light is close to that of the mercury e line, 546.07 nm which is used as a representative wavelength for visible light. Red light requires a longer photodiode, and, thus, a conservative design. Therefore, the green light and red light are used in the following analyses.

On the basis of a 120-nm process, fundamental shape parameters are determined as in Table 1. The layout of the front side shown in Figure 1b was drawn first based on our design experience and adjusted in a preliminary design with potential and Monte Carlo (MC) simulations. The pixel size is 11.4 μm × 11.4 μm. The design has an empty space beside a couple of reset drains. If these spaces are used for one drain for neighboring two FDs, the pixel size is reduced to about 10 μm. Furthermore, a checkerboard pixel arrangement rotated by 45 degrees can reduce the pixel pitch to 7.07 μm (= 10/$\sqrt{2}$
μm). If the imaging area is 11.4 mm × 11.4 mm and the pixel pitch is 11.4 μm, the pixel count reaches 1,000,000.

The size of the guide pipe is 5 × 5$\;{\mu m}^{2}$, and, thus, the fill factor is 19.2%. Therefore, an on-chip micro-lens is necessary. In the case, the direction of light is inclined toward the center. However, in the simulations, the light is assumed to be perpendicularly incident to the backside for a conservative design.

The total thickness of the sensor is assumed to be (3δ+0.1 μm). The thickness of 3δ is for the vertical length of the photodiode, in which 95% of incident light is converted to electrons. The thickness of 0.1 μm is added for a p^+^ (boron) layer for the backside hole accumulation layer. The total thickness was allocated to the length of the guide pipe and the thickness of the circuit layer.

The total thickness is 5.3 μm for green light and 12.1 μm for red light. Hereafter, the models with the total thicknesses are named Model M with a medium-sized photodiode, and Model L with a long photodiode.

#### 2.3.3. Design Targets

Table 2 shows the performance parameters aimed in the design. Since this paper shows a feasibility study of the branching image sensor, the acceptable values of the parameters are shown as well as the theoretical limits or ideal values. The values in the table are estimated, based on the current technologies and the trend in the scientific field as follows:
(1)Temporal resolution: the theoretical limit for silicon image sensors is 11.1 ps; however, as the existing highest temporal resolution of image sensors for visible light is about 10 ns, the resolution of 1 ns may be acceptable for a provisional target; in the case, the readout rate from each FD to a stacked memory chip is 6 ns (=1/12 × 2) for a correlated double sampling (CDS) operation, which may be achieved in the near future,(2)Frame count: fifty frames enable a replay of images at 10 fps for 5 s; the replay at 10 fps looks smoothly due to a saccade motion of our eyes with the lowest frequency less than 10 Hz; the duration of 5 s is at least necessary to activate dynamic recognition of our eyes; on the other hand, as televisions replay images at 30 fps (or 25 fps), a sequence of 300 frames enables replay for 10 (12) seconds for a standard TV,(3)Pixel count: the acceptable level may be 256 × 256 pixels (65,536 pixels), which is rounded to 100,000 pixels; the pixel count for 1000 × 1000 is sufficient for most scientific researches,(4)E-field: the critical field, 25 kV/cm, is ideal, at which the drift velocity is 95% of the saturation drift velocity as shown in Figure 5 [9]; up to about 100 kV/cm, the drift velocity is almost constant, and dark current increases little, because excessive photon energy is mostly converted to a lattice vibration (generation of phonon or heat); on the other hand, at 5 kV/cm, the drift velocity reduces only to the half of the saturation drift velocity.(5)Other performance parameters: 10% may be acceptable.

Some applications of ultra-high-speed imaging require much higher performance levels than the ideal levels in Table 2. For example, an ideal frame count is 4000 frames for an imaging TOF MS (time-of-flight mass spectrometry). In astronomy, an image sensor with one billion pixels is observing a wide sky. The transfer rate per readout signal line of the sensor is almost the same as that in advanced continuous-readout high-speed image sensors.

### 2.4. Destinations of Photo-Electrons

Most electrons generated by incident light directly move to the collecting FD_C_ to create a signal electron packet, but others travel to different waiting FDws. Table 3 shows the destinations of generated electrons with their symbols.

In the guide pipe, some electrons collide with the wall or return to the backside due to the random motion. They are reflected or trapped at the wall. Trapped ones are released later or recombined with holes. The reflected ones move as normal electrons again. It is difficult to estimate the reflection ratio. Therefore, it is tentatively assumed that the reflection ratio is zero, i.e., electrons once colliding against the wall or the backside disappear. On the front side, most electrons surviving collisions are collected by the collecting gate. Others migrate to FD_W_s or to neighboring pixels.

For example, the collection rate $R_{C}\;$and the migration rate $R_{mg}\;$denote the portions of electrons arriving at a FD_C_ and FD_W_s. The sum of the six factors in Table 4 is unity.

## 3. Potential Simulations

### 3.1. Guide Pipe

Hereafter, the sensor performance at the pixel level is optimized separately for the guide pipe and for the circuit layer. Later, the overall performance will be evaluated. This section discusses the guide pipe design and performance.

The average vertical E-field at the center of the guide pipe is fixed at 25 kV/cm. Then, the standard deviation of the travel time of electrons inside the guide pipe depends on the length $L_{GP}$, the width $W_{GP}$, and the concentration $p_{GP}$ of boron ions implanted in thin layers over the walls of the guide pipe. A narrow and tall guide pipe increases the collision probability of electrons to the wall. The higher boron concentration forces electrons to move toward the center, thus decreasing the collision probability, and, yet increasing the arrival time.

Figure 6 shows the potential contours in cross-sections for (**a**) Model M, and (**b**) Model L for different boron concentrations from 0 to $4\times 10^{16}\;{\mathrm{cm}}^{-3}$. Figure 7 shows the vertical potential profiles at the center of the pixel. The channel potential value is shown at the highest value close to the surface where the line is crooked.

Figure 6 and Figure 7 show that:(1)with no wall boron doping, the contours in the guide pipe are parallel, and the vertical potential profile is almost linear with the field close to the average 25 kV/cm,(2)with the boron doping, the contours near the walls bend upward so that the field is directed to the center to reduce the collision probability of electrons; the vertical potentials bend upward, decreasing the fields in the backside layer, and increasing those in the front-side layer,(3)for Model M with a high boron concentration of $4.0\times 10^{16}\;{\mathrm{cm}}^{-3}$, the fields in the backside layer and the front-side layer are respectively15 kV/cm and 34.4 kV/cm which are still within the acceptable range shown in Table 2,(4)for Model L for the same boron concentration, the field in the backside layer is 1.6 kV/cm, which is too small, resulting in a high backside returning rate, and the field in the front-side layer 79.3 kV/cm, which is significantly large, resulting in a slight increase of dark current, and(5)a higher boron concentration for Model L loops the contours in the circuit layer as shown in circles in Figure 6(b-2),(b-3), hampering delivery of signal electrons to the storage areas, which requires detailed modifications in the design.


Figure 8a,b shows the collision rates $R_{col}\;$and the standard deviations σ of the arrival time respectively for green incident light and red incident light. The standard deviation is evaluated in the reach $L_{PG}\;$from the surface and the bottom of the guide pipe. The figures show that:
(1)the collision rate $R_{col}\;$efficiently decreases for the wall boron concentration$\;p_{GP\;}$from 0 to $1.0\times 10^{16}\;{\mathrm{cm}}^{-3}$, and slightly reduces and stays less than 8% for both green and red light, which satisfies the acceptable level of 10%,(2)for Model M, the standard deviations σ is less than 20.0 ps; for Model L, σ is less than 30 ps for $p_{GP}<1.0\times 10^{16}\;{\mathrm{cm}}^{-3}$, but, suddenly increases to more than 200 ps at $p_{GP}=4.0\times 10^{16}\;{\mathrm{cm}}^{-3}$, and(3)therefore, the optimum boron doping on the wall of the guide pipe is $p_{GP}=1.0\times 10^{16}\;{\mathrm{cm}}^{-3}$.

Table 4 summarizes the collision rate and the temporal resolutions in the optimum case.

### 3.2. Branching Channels

#### 3.2.1. Modulation Functions

The voltages applied to gates are estimated by using modulation functions, i.e., the relations between the gate voltages and the channel potentials at the gates. A systematic procedure to estimate the gate voltages is proposed. Model L is used in the following explanation. The same procedure can be applied to the gate voltages of Model M.

Figure 9a,b shows the positions where the modulation functions are evaluated, which are the centers of GG, FG, and SG. The pinning gate voltages are about − 6 V. When the gate voltages are above the pinning condition, the modulation functions linearly increase as well known. The modulation factors are 0.88 for GG, and 0.82 for FG, decreasing to 0.65 at SG due to the narrow channel effect. Pinning voltages and the corresponding potential values for the gates slightly change due to the narrow channel effect for the same dopant condition over the whole area. Table 5 summarizes the modulation functions.

#### 3.2.2. Gate Voltages and Potentials of the Channels

Figure 10a conceptually shows an ideal channel potential profile from FDw to FDc. The ideal horizontal potential profile linearly increases from an upstream (low-voltage) SG toward a downstream (high-voltage) SG. From the channel potential value and the modulation function at each gate, the gate voltage is uniquely decided as follows:(1)assume the amplitude of the SG voltage is 3 V, and the pinning voltage is −6 V, and, then, the low and high voltages of the SG are −6 V–3 V,(2)from the modulation function of SG, the channel potentials at an upstream SG and a downstream SG are −3.73 V and −1.78 V,(3)the gate voltages of GG and upstream and downstream FGs are interpolated from Figure 10a as −5.19 V and −3.84 V, and, then, the amplitude of FG is 1.35 V.

In Table 6, and Figure 10b, the estimated potential values are compared with those calculated by a potential simulation with the gate voltages. The errors are 0.9% to 13.8%, which proves a high efficiency of the proposed simple estimation method of the gate voltages. 

The voltage amplitude of FG, 1.35 V, may be allowable for semiconductor circuits driven at a very high frequency. The voltage amplitude of SG, 3 V, is higher than that of FG. However, the operation rate of SG can be 1/6 or 1/12 of that of FG, and the gate area of one SG is only 45% (1.14$\;{\mu m}^{2}$/2.49$\;{\mu m}^{2}$) of that of FG. Therefore, if FG can be driven at 1.35 V amplitude, SG with a 3 V amplitude can be easily operated.

The average field is 3.05 kV/cm (=1.95 V/6.38 μm).

### 3.3. Local Potential Adjustment 

#### 3.3.1. Voltage Adjustment

Figure 10b shows the following problems:(1)the average field is 3.05 kV/cm as calculated from the potential line in the figure, which is about 1/10 of the ideal field, 25 kV/cm, which reduces the drift velocity of electrons to 1/3 of the saturation drift velocity as shown in Figure 5,(2)the potential profiles at the centers of the gates are rather flat, further decreasing the field and increasing the travel time, and(3)at the downstream FG, there is a slight potential dip as shown with a red circle, which seriously increases the travel time of electrons passing through the area.

On the other hand, the figure provides suggestions for modifications of the potential profile as follows:
(1)a fine semiconductor process: if the same voltage amplitude is applied, a fine process increases the average field inversely proportional to the pixel size, and, in addition, portions of a steep fringe field erode the flat portions, and(2)voltage modifications: if the potentials at the downstream FG is slightly lowered, and SG is raised, the small potential dip may disappear; for the (−0.1 V) decrease for the FG and 0.3 V increase for the SG, the modulation functions suggest that gate voltages must be shifted by −0.12 V (=−0.1/0.82 V) for the FG, and 0.46 V (=0.3/0.65 V) for the SG.


Then, the potential dip disappears. The voltage amplitudes of the FG and the SG are slightly modified to 1.2 V and 3.5 V.

#### 3.3.2. Detailed Adjustment of Temporal Resolution for Potential Barrier

As shown in Figure 3, there are three equivalent electron routes along the upstream SGs to GG; R_A_, R_B_ and R_C_, and three equivalent routes from GG to downstream FD_C_; R_D_, R_E_ and R_F_. As shown in Figure 10b, the field around the upstream SG and FG is high, creating a relatively high drift velocity. On the other hand, the field along the routes in the downstream FG is relatively low even after adjustment to eliminate the slight potential dip. The downstream route is represented by one of the equivalent routes, R_D_, R_E_ and R_F_, in Figure 3. The potential profiles along these routes are slightly different around zero, which causes a large difference in travel times of electrons passing through these routes, and seriously unbalances the arrival times of electrons depending of the routes.

This may seem a local problem. The important role of this paper is not only to show general aspects, but also to find these potential risks which may cause serious problems in development of the prototype and to provide ways to solve them. Therefore, the travel times for the routes R_D_, R_E_ and R_F_ are carefully evaluated to be shortened and balanced by MC simulations. The position of the electron source is at 1 μm above the front surface at the center of the pixel to separately evaluate the performance in the circuit layer from that in the guide pipe. Table 7 shows the results.

The ratio of the average arrival times of 231 ps and 838 ps of the routes R_D_ and R_E_ ps is 27.6%. Figure 11(a-1) shows the plane potential profiles of the channels and Figure 11(a-2) shows the potential profiles along the routes. The route R_E_ shows a small dip between the downstream FG to SG, which caused the large difference in the temporal resolutions.

To mitigate the unbalance in the travel times, fine modifications are applied to the design of a phosphorus mask as shown in Figure 11b. The improved potential profiles on the circuit layer, and the improved travel times are shown in Figure 11(c-2) and Table 7.

Table 7 shows that the travel times from the center of the pixel on the front side to the FDs are 212 ps to 231 ps, which improve the ratio of the balance to 91.8%. 

## 4. Overall Evaluation

### 4.1. Fundamental Performance

The overall performance is evaluated for the combined structure with the guide pipe and the circuit layer. Green light represents visible light. In a theoretical analysis, it is appropriate to use Model M for the corresponding wavelength. However, red light incident to Model M in Table 4 cannot be fully absorbed and generates false signal electrons in the circuit layer. Therefore, Model L with incident green light is selected for safe-side evaluation in practice. Electrons are generated by incident light to the backside. The results are summarized in Table 8, showing:(1)all major performance indices satisfy the acceptable values listed in Table 2; except the frame count,(2)for example, the temporal resolution is about 386 ps, which is much shorter than the acceptable level, 1 ns;(3)about 88% of electrons are collected by a collecting FD_C_; actually, some portions of electrons colliding against the walls of the guide pipe and the backside are reflected and join the collected electrons; therefore, the real collection rate increases to more than 90%.

The frame count is 12 for the burst imaging, and can be increased to 48 for a macro-pixel operation with four neighboring elemental pixels, sacrificing the sensitivity to 1/4. In the near future, the continuous imaging can be introduced by transferring image signals from 12 FDs in turn to a stacked memory chip, keeping a reasonable noise level by reducing the transfer rate with the multiple FDs.

### 4.2. Visual Diagonosis with MC Simulations

Figure 12 shows example trajectories of the signal electrons generated with MC simulations for the modified design of Model L and green light of 550 nm. Light is perpendicularly applied to the backside. The figures show that:(1)as shown in Figure 12a, most of signal electrons generated by incident light forms a bundle, which is effectively squeezed to the center of guide pipe to minimize the collision rate,(2)most of them fall onto the guide gate and transferred to FD_C_,(3)however, some electrons take extraordinary long travel times to increase the temporal resolution, as shown with A and B, and(4)few of them migrate to FD_W_ as shown with C.

The example trajectory A is caused by electrons generated and wandering in a low-field layer created by a thin boron doped layer at the backside. A and B are caught at a local no field on the front side. The problem at the backside layer may be mitigated with a negatively charged thin layer of Al_2_O_3_ over the backside insulation layer replacing the thin boron layer. In the simulation, the direction of light is perpendicular to the backside. An on-chip micro-lens may efficiently suppress the collision against the wall and the migration to FD_W_s. MC simulations are very useful to visually diagnose local problems and modify the design.

### 4.3. Further Confirmation and Evolution

#### 4.3.1. Metal Wiring and 3D Stacking of Driver Chip

This paper has mainly discussed a potential design and temporal resolution analyses. There are many other factors relating to the feasibility of an image sensor developed with these new concepts. For Hanabi, a large number of metal wires increases the chip area. A narrow contact pitch to stack a driver chip on the sensor chip seriously decreases yield.

Burst imaging with Hanabi requires at least 34 metal contacts, including 19 (= 1+6+12) gate electrodes and 12 signal lines from the FDs. From our experience, the minimum pixel size restricted by the number of wires is approximately estimated by (the process node) × 3 × (the number of metal contacts). For the current layout with a 120-nm process, the product makes 12.24 μm, which is close to 11.4 μm of the pixel size shown in Table 8. Therefore, the number of wires does not seriously restrict the feasibility of the burst imaging mode, but may require a slightly large pixel. More wires are necessary to expand the concept to the continuous imaging Hanabi, which requires a larger pixel size or a finer process.

The driver circuit is installed in a stacked bottom tier. A pixel of Hanabi has 21 high-speed gate electrodes. Therefore, the pitch of the Cu-Cu contacts in each pixel is 2.49 $\mu m\;\left(=11.4/\sqrt{21}\right)$, which can be processed with the existing technology. Additional discussions in this section have further confirmed the overall feasibility of Hanabi.

#### 4.3.2. Higher Frame Rate: Toward Super Temporal Resolution

Theoretically, Hanabi can achieve noiseless imaging. The pixel count and the frame rate can be increased by downward scaling. The pixel count increases in inverse proportion to the square of the pixel size. Further increase of the frame rate is discussed here in relation to the scaling.

The standard deviation $\sigma_{T}$ of the total arrival time from the generation site in the guide pipe to FD_C_ is expressed as follows:(1)$$\sigma_{T}^{2}=\sigma_{GP}^{2}+\sigma_{CL}^{2}$$
where $\sigma_{GP}$ and $\sigma_{CL}$ are the contributions of the arrival times due to motions of signal electrons, in the guide pipe and in the circuit layer, respectively. The values of $\sigma_{T}$ and $\sigma_{GP}$ are shown in Table 8 and Table 4 to be 125 ps and 29.2 ps, respectively. By substituting these values in (1), $\sigma_{CL}$= 121.5 ps. Therefore, the temporal resolution is almost solely determined by the horizontal motion on the front side. The value is slightly larger than that for electrons starting at the pixel center, 113 ps, shown in Table 7.

Further analysis is needed to eventually suppress the horizontal motion on the circuit layer increasing the total temporal resolution. There are two major causes of horizontal motion:(1)as shown in Figure 10b, the horizontal potential profile on the front side consists of flat parts in the middle of the gates, and high-field parts at the gate boudaries due to the fringing effect; the flat parts seriously elongate the standard deviation of the arrival time, and(2)as shown in Figure 12a, most electrons fall on the guide gate, and move horizontally; the different horizontal travel distance over the guide gate causes horizontal mixing.

Downward scaling mitigates both problems as follows:(1)for shorter gates, the high-field portions erode the flat parts, linealizing the whole horizontal potential profile, and(2)for a narrower guide gate, the horizontal mixing effect reduces in proportion to the size.

Potential simulations show that the horizontal potential profile becomes almost linear with no flat portion for a 1/3 model of the current one. Then, the 1.65-μm guide gate in the current layout shrinks by about 0.5-μm. Then, the standard deviation $\sigma_{CL}$ is calculated by the MC simulation for signal electrons uniformly distributed all over the guide gate. The result is as follows:
(1)$\sigma_{CL}=8.5\;\mathrm{ps}<\sigma_{GP}=$29.2 ps (the standard deviation in the upper silicon guide pipe).(2)the theoretical temporal resolution for the horizontal motion on the front side is $2\sigma_{CL}=17.0\;\mathrm{ps},$ which is close to the theoretical temporal resolution limit of the silicon photodiode 11.1 ps [8].

Most image sensors are based on silicon technology. Therefore, we named imaging technology that overcomes the silicon limit on the frame interval, ‘the super-temporal-resolution (STR) imaging’. The STR Hanabi is within the scope of realistic research by replacing the photodiode material from silicon to germanium [9].

Germanium has some fundamental problems in the context of a possible integration in the Hanabi technology, such as a very high dark current. However, the dark current generated in 11.1 ps is 1/10^11^ of the dark current generated in one second. Other problems, including the instability of the oxide for the insulation layer, are currently being addressed [23,24,25].

## 5. Conclusions

A burst image sensor is a standard technology for ultra-highspeed imaging, using either CCD or CMOS technologies. A BSI branching image sensor with a stacked driver chip is proposed to find a compromise between their pros and cons. A pixel of the sensor is composed of an upper photoelectron conversion layer to converge generated charges to the center, and a lower circuit layer to cascade the charges through radially branching channels to multiple storage units. Since such a sensor structure has not existed nor been tested, the technical feasibility of the sensor is confirmed with simulations and additional considerations.

First, a design strategy specific to the sensor structure is proposed and applied to a model case. The upper photoelectron conversion and the lower circuit layers are separately analyzed to solve associated problems, and, then, the overall performance is evaluated for the whole sensor structure.

Under very conservative design conditions with cooling, the sensor achieves theoretically noiseless imaging with the frame interval of less than 413 ps and the frame count of 12 to 48. In the near future, the frame count can be increased with a CMOS readout operation to a stacked memory chip, sacrificing the noiseless image capture.

## Figures and Tables

**Figure 1 sensors-21-02506-f001:**
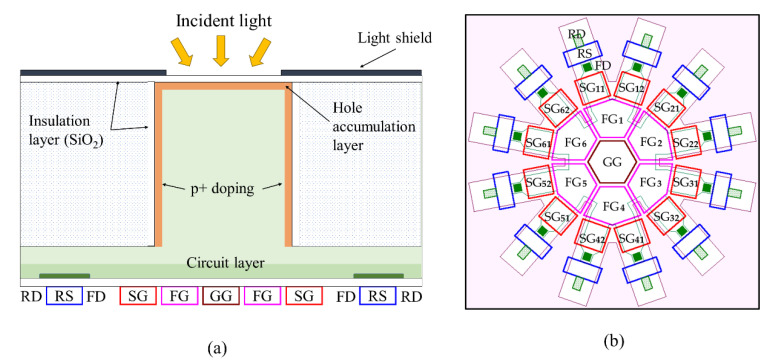
A BSI CCD/CMOS hybrid branching image sensor, Hanabi: (**a**) a cross section, (**b**) elements on the front side (GG, guide gate; FG, first-branching gate; SG, second-branching gate; FD, floating diffusion; RS, reset gate; RD, reset drain).

**Figure 2 sensors-21-02506-f002:**
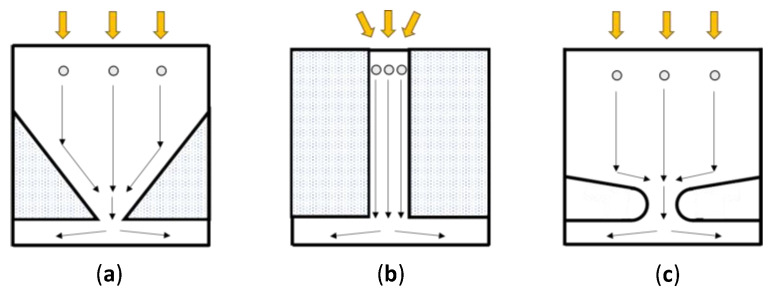
Technologies proposed for collection of signal electrons to the pixel center and potential separation of photodiode in the upper layer and the circuit in the lower layer [5]: (**a**) charge collection pyramid, (**b**) light/charge guide pipe, and (**c**) p-well; the guide pipe and the pyramid significantly increase the frame rate.

**Figure 3 sensors-21-02506-f003:**
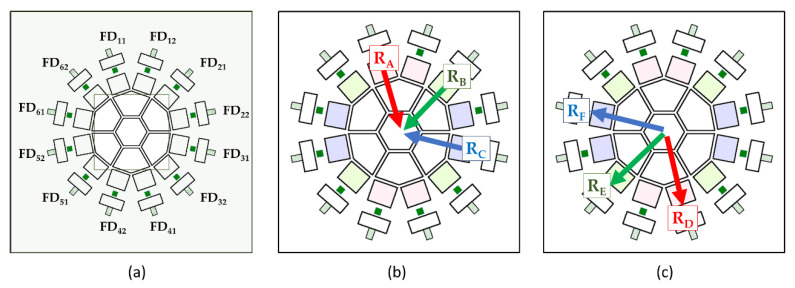
Equivalent travel routes of signal electrons: (**a**) positions of FDs to show gate numbering, (**b**) three representative routes from an upstream (low-voltage) SGs to GG, coded with three colors, (**c**) three representative routes from GG to an FDc.

**Figure 4 sensors-21-02506-f004:**
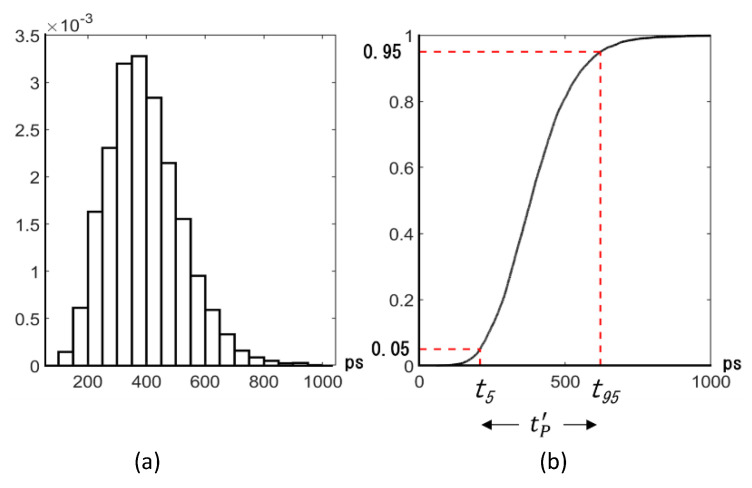
Definition of the practical temporal resolution $t_{P}^{\prime }$: (**a**) probability density function, (**b**) cumulative distribution function, $t_{P}^{\prime }=t_{95}-t_{5}.$

**Figure 5 sensors-21-02506-f005:**
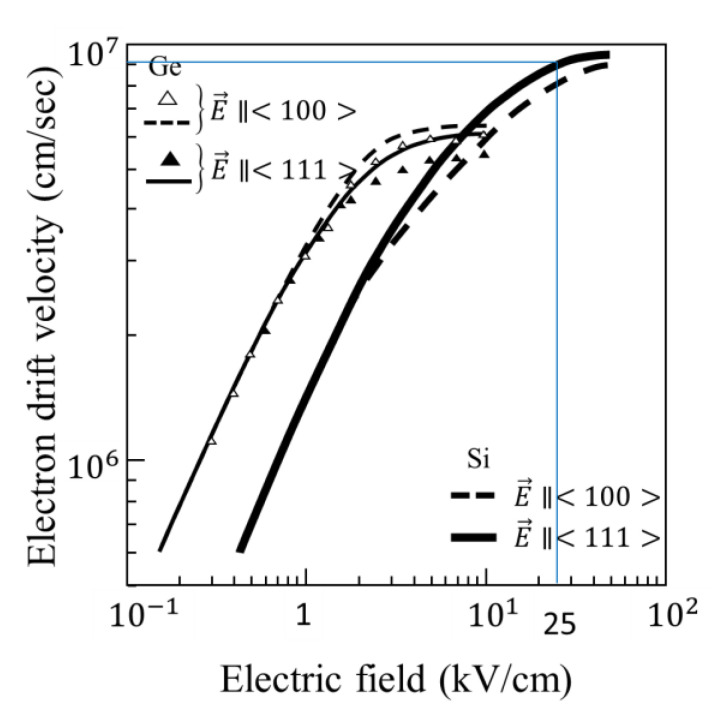
Electron drift velocity for intrinsic silicon and germanium (adopted with permissions respectively from ref. [21] by Elsevier on 11 February 2021, and ref. [22] by APS on 29 March 2021.

**Figure 6 sensors-21-02506-f006:**
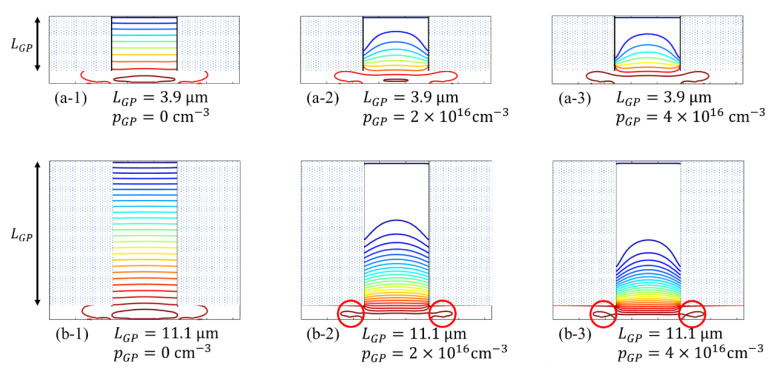
Potential contour lines for (**a**) Model M and (**b**) Model L, where $L_{PG}$ and $p_{GP}$ are the length and the concentration of the boron layers over the walls of the guide pipe.

**Figure 7 sensors-21-02506-f007:**
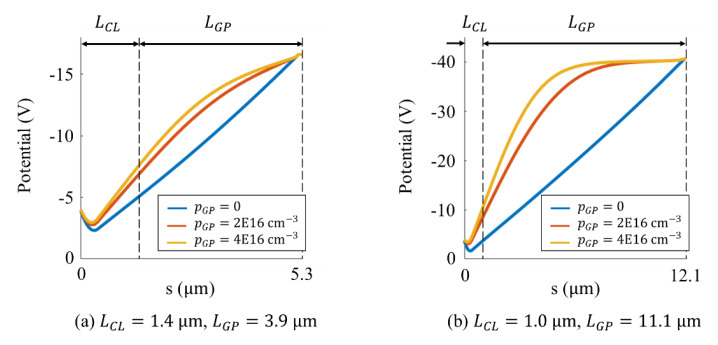
Vertical potential profiles at the center of the pixel: (**a**) Model M (L=5.3 μm) and (**b**) Model M (L=12.1 μm); s is the distance from the front surface; $p_{GP}\;$is the boron concentration over the walls of guide pipe; $L_{PG}$ and $L_{CL}$ are the length of the guide pipe and the thickness of the circuit layer; $L=L_{PG}+L_{CL}$.

**Figure 8 sensors-21-02506-f008:**
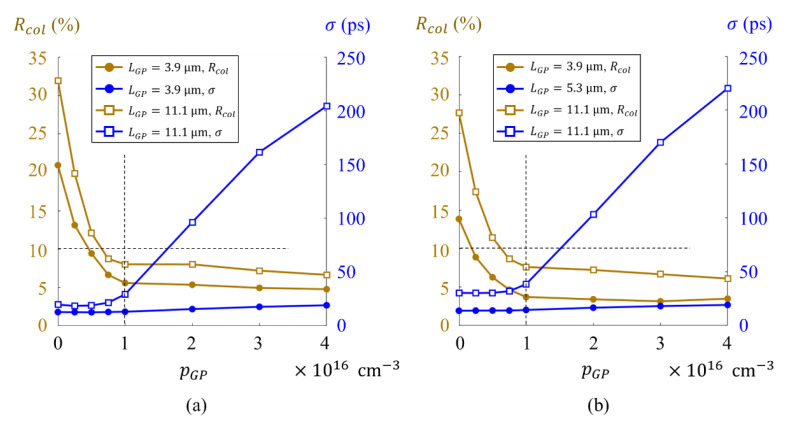
The collision rates $R_{col}$ and the standard deviations σ of the arrival time for the wall boron concentration $p_{GP}:\;$blue curves, the standard deviations of arrival times to the bottom of the guide pipe; brown curves, the collision rates; lines with circles and squares are for Model M and Model L: (**a**) green incident light (550 nm), (**b**) red incident light (650 nm).

**Figure 9 sensors-21-02506-f009:**
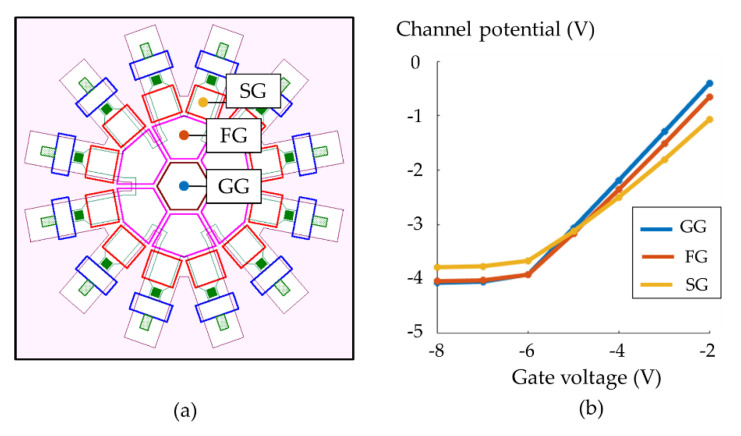
Modulation functions (channel potentials vs. gate voltages): (**a**) positions; centers of GG, FG, and SG, at which the relations are evaluated, (**b**) blue, red, and yellow curves show modulation functions at GG, FG, and SG.

**Figure 10 sensors-21-02506-f010:**
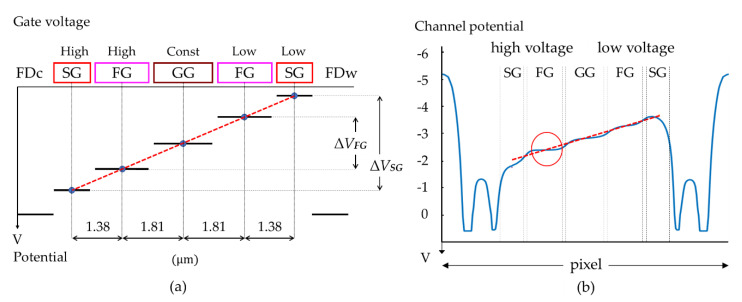
Potential profiles to transfer electrons from an upstream (low-voltage) SG to a collecting FDc: (**a**) a conceptual explanation to estimate gate voltages; (**b**) potential profiles calculated with the proposed systematic estimation procedure; dashed red lines are the ideal linear potential profiles; $\Delta V_{FG}\;\mathrm{and}\;\Delta V_{SG}$ are respectively the potential amplitudes at FG and SG; there is a slight dip near the downstream (high-voltage) SG shown with a red circle.

**Figure 11 sensors-21-02506-f011:**
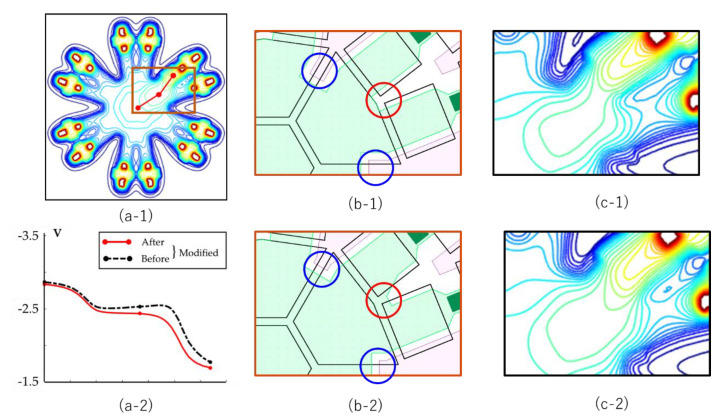
An example of local potential adjustments from GG to FD_21_ (downstream green route R_E_ in Figure 3**c**): (**a-1**), a 2D planar channel potential on the front side before the adjustment, and a route on which the potential is evaluated; (**a-2**): the potentials before and after the modification; (**b-1**,**b-2**), phosphorus masks shown with green before and after the modification; (**c-1**,**c-2**), enlarged figures of the local potentials.

**Figure 12 sensors-21-02506-f012:**
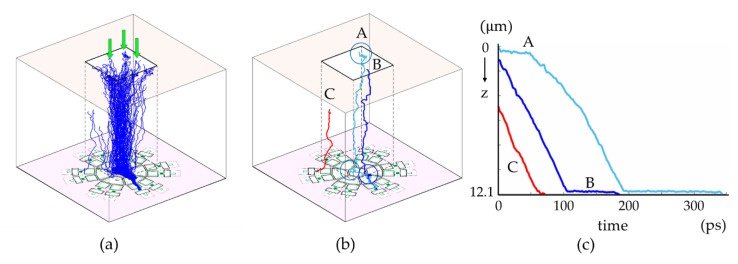
Trajectories of signal electrons generated with MC simulations for the modified design of Model L and green light of 550 nm: (**a**) trajectories of signal electrons, (**b**) examples of unusual trajectories; A and B take extraordinary long travel times, C moves to a waiting FD_w_, (**c**) travel times of A, B and C vs. travel distances from the generation sites to FDs.

**Table 1 sensors-21-02506-t001:** Representative wavelengths of light and fundamental shape parameters: * the total thickness is (3δ+0.1 μm).

Color	Model	Wavelength λ (μm)	Penetration Depth δ (μm)	Thickness (μm)		
				Total *	Guide Pipe	Circuit Layer
Blue		450	0.49			
Green	M	550	1.73	5.3	3.9	1.4
Red	L	650	4.00	12.1	11.1	1.0
Pixel size and pitch				11.4 μm × 11.4 μm		
Guide pipe width/Fill factor				5 μm × 5 μm/19.2%		
Thickness/concentration of p^+^ layer over the guide pipe				0.1 μm/(0–4$)\times 10^{16}$cm^−3^		

**Table 2 sensors-21-02506-t002:** Performance parameters aimed in the design in this paper: * an on-chip micro-lens array is necessary.

Parameters	Target	
	Acceptable Level	Ideal Level
Temporal resolution	1 ns	11.1 ps
(Frame rate)	(1 Gfps)	(90.1 Gfps)
Frame count	50	300
Pixel count	100,000	1,000,000
E-field	5 kV/cm to 100 kV/cm	25 kV/cm
Temporal crosstalk	10%	0%
Spatial crosstalk	10%	0%
Fill factor	10% *	100%
Collision rate to guide pipe	10%	0%
Backside returning rate	10%	0%

**Table 3 sensors-21-02506-t003:** Destinations of signal electrons.

Factors	Symbols	Destinations
Collection rate	*R _c_*	collecting FD_C_
Migration rate	*R _mg_*	waiting FD_W_s
Backside returning rate	*R _back_*	backside
Collision rate	*R _col_*	walls of the guide pipe
Waste rate	*R_w_*	passing through the front side
Spatial crosstalk rate	*R_sc_*	neighboring pixels

**Table 4 sensors-21-02506-t004:** Collision rates and temporal resolutions in the optimum case: $p_{GP}=1.0\times 10^{16}\;{\mathrm{cm}}^{-3}$ * the standard deviation used in Section 4.3.2.

Incident Light		Model		Standard Deviation of Arrival Time σ (ps)	Practical Temporal Resolution $t_{P}=3.3\sigma \left(ps\right)$	Collision Rate to Guide Gate Walls (%)
Wavelength λ (nm)		**Guide Pipe Length** (μm)				
Green	550	M	3.9	13.0	42.9	5.6
		L	11.1	29.2 *	96.4	8.0
Red	650	M	3.9	14.3	47.2	3.7
		L	11.1	38.3	126.4	7.6

**Table 5 sensors-21-02506-t005:** Modulation functions along the center line$:\;V_{XX}$ and $P_{XX}$ are the gate votage and the channel potential of the gate *XX*; the pinning voltage is about −6 V.

Position	Modulation Function (V)	Pinned Channel Potential (V)
GG	$P_{GG}=0.88V_{GG}+1.35$	−3.93
FG	$P_{FG}=0.82V_{FG}+0.95$	−3.95
SG	$P_{SG}=0.65V_{SG}+0.17$	−3.73

**Table 6 sensors-21-02506-t006:** Gate voltages and channel potentials.

	Methods	GG	FG			SG		
			Low	High	Amplitude	Low	High	Amplitude
Voltage (V)	Proposed method	−4.66	−5.19	−3.84	1.35	−6.00	−3.00	3.00
Potential (V)		−2.79	−3.30	−2.16	1.14	−3.73	−1.78	1.95
	Simulation	−2.86	−3.33	−2.50	0.83	−3.61	−1.66	1.95
Error (%)		2.71	0.91	13.8		3.28	7.10	

**Table 7 sensors-21-02506-t007:** Travel times from the center of GG to FDc before and after detailed modification of a phosphorus mask: electrons start from 1.0 μm above the front side at the center of the pixel, * the value is used in Section 4.3.2.

Temporal Criteria	Before Modification			After Modification		
	R_D_	R_E_	R_F_	R_D_	R_E_	R_F_
Average (ps)	231	838	452	222	231	212
σ (ps)	111	710	317	106	99	113 *

**Table 8 sensors-21-02506-t008:** Overall performance of the sensor: * the value is used in Section 4.3.2; ** frame count is for burst imaging, including a macro-pixel operation with two or four elemental pixels; *** a sufficiently low temperature and slow readout theoretically enable noiseless transfer of signal electrons in the Hanabi structure; ^※^ destinations of electrons and temporal indices are calculated by MC simulations for the equivalent routs, R_D_, R_E_ and R_F_, and their largest and smallest values are listed; ^※※^ for simulations, *R_w_* = (0.05–0.2) = 1 − (sum of other portions), and the exponential attenuation of incident light is calculated to be 0.09.

Fundamental Performance			
Pixel	Size & thickness		$11.4\times 11.4\times 12.1\;{\mu m}^{3}$
Guide pipe	Width & length		$5\times 5\times 11.1\;{\mu m}^{3}$
Incident light: green			550 nm
Frame rate *			2.59 Gfps
Frame count **			12–48
Pixel count			1,000,000
Noise ***			Very small
Fill factor			19.2%
E-field		Guide pipe	25 kV/cm
		Circuit layer	2.4 kV/cm
Destinations of electrons ^※^ (%)FIGURE		*R_c_*	87.8–88.5
		*R_mg_*	0.11–0.12
		*R_col_*	8.12–8.85
		*R_back_*	3.05–3.26
		*R_w_*	0.05–0.2 (0.09) ^※※^
		*R_sc_*	0
Temporal indices ^※^ (ps)		Mean	322–340
		σ	109–125 *
		$t_{P}$	360–413

## Data Availability

Not applicable.

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
