# Peer review of "A Pixel Design of a Branching Ultra-Highspeed Image Sensor"

_sensors, 2021, doi:10.3390/s21072506_

Round 1
Reviewer 1 Report
Review: A Branching Ultra‐highspeed Image Sensor
From my point of view, the work has analyzed the design of pixel for ultra-high speed image sensor “Hanabi” in detail, but there are still many shortcomings. Just, I suggest major modifications:
Comments:
- Pg3 line 101: “Second‐branching gates connected to F1…”, The F1 may be FG1.
- Pg3 line 109: “…RD, resent drain”, Please confirm the spelling.
- In Pg4 figure 2, it is not clear to this reviewer that what is the functional difference between the potential distribution in figure2 (a) and figure2 (c)? According to figure1 (a), what are the formation conditions of these two potential distributions. Please give some brief explanations.
- In Pg4 section 2.1.2, the authors state that “…from eleven low‐voltage (upstream) SGs toward a high‐voltage (downstream) SG…”, but it is not clear to this reviewer where the charge of SG come from and why the charge transfers from one SG to another. The authors mean that SG is also the photoelectric conversion region or SG is the high potential region during exposure, and the charge at GG will move to each SG? What is the purpose of designing charge transfer between SGs?
- Pg6 line 240: “4 μm x 11.04 μm”, Please confirm the pixel pitch.
- In Pg7 section 2.3.3, the author lists many estimated model parameters, among which the temporal resolution set at 1ns is far beyond the current limit value. Could the author give some analysis or basis?
- In section 4.1 paragraph 1, the authors used Model L and green light, but Model L is long photodiode for red light (in line 253-255). Why?
- In Table 8, noise is 0 and in line 536, the authors state that “keeping a reasonable noise level by reducing the transfer rate with the multiple FDs”. Can authors do some analysis about noise and transfer rate?
- In figure 10(b), the reviewer understands that the SG on the right side connected to the FD does not need to be read and the left side is not. In figure 1(a), SG is under light shield. Why did the authors set it to the middle voltage and not the lowest voltage? Does the authors think that setting the lowest voltage will result in lower charge leakage?
- In line 584, it is said that “Hanabi can achieve noiseless imaging”. It is not clear for the reviewer why noiseless imaging?
- The title of the manuscript is “A Branching Ultra‐highspeed Image Sensor”, but there is a lot of pixel analysis and a little bit of circuit structure in this article. The reviewer thinks the title of the manuscript should be “Pixel”, not “Sensor”.

Reviewer 2 Report
-Performance comparison
Performance comparison to the state-of-the-art will be beneficial for benchmarking the work.
Reviewer 3 Report
Pleasee see the attached file

Round 2
Reviewer 1 Report
Thank you for your answers to the comments, which helped me understand the parts in the manuscript that I did not understand.
After the insertion of some explanatory content, the organization and hierarchy are clear and distinct. The full manuscript is comprehensive and easy to understand. In my opinion, the paper is largely written well. Just, I suggest accept in present form.
Reviewer 3 Report
Authors have answered all the questions, and I don't have any comments anymore.